# A Comprehensive Analysis of Cutaneous Melanoma Patients in Greece Based on Multi-Omic Data

**DOI:** 10.3390/cancers15030815

**Published:** 2023-01-28

**Authors:** Georgia Kontogianni, Konstantinos Voutetakis, Georgia Piroti, Katerina Kypreou, Irene Stefanaki, Efstathios Iason Vlachavas, Eleftherios Pilalis, Alexander Stratigos, Aristotelis Chatziioannou, Olga Papadodima

**Affiliations:** 1Institute of Chemical Biology, National Hellenic Research Foundation, 11635 Athens, Greece; 2Centre of Systems Biology, Biomedical Research Foundation of the Academy of Athens, 11527 Athens, Greece; 31st Department of Dermatology, Andreas Syggros Hospital, Medical School, National and Kapodistrian University of Athens, 16121 Athens, Greece; 4Division of Molecular Genome Analysis, German Cancer Research Centre, 69120 Heidelberg, Germany; 5e-NIOS Applications Private Company, 17671 Kallithea, Greece

**Keywords:** cutaneous melanoma, whole exome sequencing, RNA sequencing, somatic mutations, SNPs, FFPE, bioinformatics

## Abstract

**Simple Summary:**

Cutaneous melanoma (CM) accounts for the majority of skin cancer-related deaths and, during recent decades, its incidence has increased worldwide. Although early-stage disease is associated with favourable prognosis, a subgroup of tumours is characterised by increased aggressiveness. High inter- and intra-tumour heterogeneity could contribute to the establishment of such aggressive phenotypes, which are based both on the genomic background and the transcriptomic plasticity of melanoma cells. Molecular profiling of melanomas has expanded our knowledge about mechanisms related to disease onset and progression, and it is key to improving optimal management. In this study, we aimed to characterise patients with primary CM for both germline and somatic variations as well as alterations at the gene expression level, by applying whole exome, targeted, and transcriptomic approaches and integrative, multi-layered bioinformatics analysis. Oncogenic alterations were identified, and distinct phenotypic cell states could be inferred from transcriptomic data.

**Abstract:**

Cutaneous melanoma (CM) is the most aggressive type of skin cancer, and it is characterised by high mutational load and heterogeneity. In this study, we aimed to analyse the genomic and transcriptomic profile of primary melanomas from forty-six Formalin-Fixed, Paraffin-Embedded (FFPE) tissues from Greek patients. Molecular analysis for both germline and somatic variations was performed in genomic DNA from peripheral blood and melanoma samples, respectively, exploiting whole exome and targeted sequencing, and transcriptomic analysis. Detailed clinicopathological data were also included in our analyses and previously reported associations with specific mutations were recognised. Most analysed samples (43/46) were found to harbour at least one clinically actionable somatic variant. A subset of samples was profiled at the transcriptomic level, and it was shown that specific melanoma phenotypic states could be inferred from bulk RNA isolated from FFPE primary melanoma tissue. Integrative bioinformatics analyses, including variant prioritisation, differential gene expression analysis, and functional and gene set enrichment analysis by group and per sample, were conducted and molecular circuits that are implicated in melanoma cell programmes were highlighted. Integration of mutational and transcriptomic data in CM characterisation could shed light on genes and pathways that support the maintenance of phenotypic states encrypted into heterogeneous primary tumours.

## 1. Introduction

Cutaneous melanoma (CM) is an aggressive cancer arising from epidermal melanocytes, which are melanin-producing cells located in the basal layer of the epidermis. Melanocytes arise from the neural crest, a transient embryonic formation consisting of highly migratory pluripotent cells, which give rise to a number of different cell types. Throughout embryonic development, melanocyte progenitors migrate, differentiate and colonise the epidermis of the skin, hair follicles, the uvea of the eye, and mucous membranes throughout the body [1,2].

CM incidence rates differ widely across countries and geographic regions; however, this rate has steadily increased in Caucasian populations worldwide in recent decades, making CM a growing epidemic [3,4]. Regarding European populations, the highest occurrence rate of CM is observed in countries of the Northern and Western Europe, such as Denmark, Sweden, and the Netherlands, while Greece belongs to the group of low-incidence countries [5,6].

Transformation of melanocytes into melanoma is a multi-factorial process involving a complex interplay of genetic and environmental risk. Exposure to ultraviolet radiation (UVR) is the most well-established environmental risk factor [7]. In addition, family history and phenotypic traits, such as blond hair, light eye colour, and tendency to freckle, as well as the number of common and atypical melanocytic nevi, are among the well-recognised melanoma risk factors [8,9]. Regarding the genetic background predisposing to melanoma, the first genetic evidence came from the identification of germline alterations in familial melanoma patients. Cyclin-dependent kinase inhibitor 2A (*CDKN2A*) was the first familial melanoma gene characterised. *CDKN2A* encodes two distinct tumour suppressors which are both involved in cell cycle regulation [10,11]. Another cell-cycle related gene found to be mutated in melanoma-prone families is Cyclin-dependent kinase 4, *CDK4*. In the case of sporadic melanomas, comprising more than 90% of all melanomas, several susceptibility loci acting as moderate (*BAP1*, *TERT*, *POT1*, *ACD*, *TERF2IP* and *MITF*) or low penetration genes have been identified [10,12,13]. Genome-wide association studies (GWASs) have also revealed numerous low-penetrant single nucleotide polymorphisms (SNPs) associated with melanoma risk in the general population with *MC1R* being one of the most prevalent ones [14,15,16,17]. Further characterisation of the genetic risk factors in different patient populations could help develop more efficient prevention strategies and improve strategies for early diagnosis.

In the last decade, Next Generation Sequencing (NGS) technologies enabled significant steps towards the characterisation of the somatic mutational profile of the melanoma genome [18,19]. It has been shown that the melanoma genome records the highest mutational load among several cancers [20], which can in part explain the high heterogeneity characterising melanoma. The landmark study from The Cancer Genome Atlas (TCGA) consortium [21] suggested the classification of CM patients into four genomic subtypes according to the most prevalent of the significantly mutated genes identified in the cohort under study. The proposed subtypes are characterised as *BRAF* mutant, *RAS* mutant, *NF1* mutant, and triple wild type (3WT). BRAF, RAS and NF1 proteins are involved in the MAPK signalling pathway, which is often deregulated in melanoma, resulting in stimulation of cell proliferation and survival.

Beyond the elevated mutational burden, intratumour heterogeneity is also supported by the increased plasticity of tumour cells and their ability to switch between different phenotypic states [22,23]. Recently, “unlocking phenotype plasticity” was proposed as a discrete hallmark capability of cancer cells [24], determining melanoma progression and therapy resistance [22]. A growing body of evidence, based on transcriptomic profiling of melanoma cell lines and tumours, suggests that melanoma cells display different transcriptional programmes that can define different phenotypic states. Two predominant transcriptional programmes were initially described that characterise a more differentiated “proliferative” state and a less differentiated “invasive” state [25,26]. Despite these well-characterised distinct programmes, it is now established that a range of transcriptional programmes exists that is governed by different transcription factors [27]. Already since the first transcriptomic studies reporting distinct transcriptional profiles of melanoma cell states, it had been suggested that a third melanoma phenotype exists, distinct from the “proliferative” and “invasive” [25]. Nowadays, it is understood that the “proliferative” and “invasive” states describe part of a series of melanoma states that represent different levels of differentiation ranging from neural-crest stem cells to pigmented melanocytes [28]. These “non-proliferative”, “non-invasive” states were often referred to collectively as “intermediate” state and include cells that are endowed with both invasive and proliferative properties [23]. The “proliferative”—also referred to as the “melanocytic”—state is characterised by medium to high expression of the melanocyte-inducing transcription factor, also known as microphthalmia-associated transcription factor (*MITF*) [29], while the “invasive”—also referred to as “undifferentiated” and “mesenchymal-like”—state is characterised by low *MITF* and high expression of *AXL* [30], a gene coding for a tyrosine kinase receptor [31], which is emerging as a key player in progression and metastasis of several cancer types [32]. Regarding the other cell states, they include the “hyperdifferentiated” state, the “intermediate” state (also mentioned as “transitory-intermediate migrating” state), the “starved” state (also mentioned as “therapy-induced starved-like” state) and the Neural Crest Stem Cell (NCSC)-like state [23,27].

Mounting evidence suggests that among melanoma cells, those expressing the neurotrophin receptor CD271 exhibit the most aggressive and invasive properties [33]. CD271—encoded by the *NGFR* gene—is a neural crest stem cell marker, [34], acting as a key regulator of the NCSC-like state in melanoma. CD271 is not only associated with stem-like properties but, acting as a molecular switch, it is involved in the regulation of melanoma migration and metastasis [35,36,37].

In this work, we aimed to characterise the mutational and transcriptomic profile of primary CMs. We previously reported the mutational characterisation of primary melanomas from patients in Greece [38]. To the best of our knowledge, no other study has since reported large-scale genomic data from Greek patients with CM. Furthermore, we compare the mutational data of this study with available data from large genomic studies and report the observed differences between BRAF V600E and V600K cases. Finally, we evaluate the transcriptomic profile of the analysed CMs for gene signatures that have been reported to capture and reflect the main cellular state of melanomas.

## 2. Materials and Methods

### 2.1. Patients

The Greek samples consisted of Formalin-Fixed Paraffin-Embedded (FFPE) tissue from patients with a histologically confirmed diagnosis of melanoma at Andreas Syggros Hospital, a large referral centre for melanoma and skin cancer in Athens, Greece. All subjects were older than 18 years of age, with a median age of 59 years. Demographic variables, pigmentation traits (eye, hair, and skin colour), skin phototype, tanning ability, and information from clinical examination were obtained through a questionnaire that was filled out by all participants under the supervision of a certified dermatologist who performed the clinical examination. The study protocol was approved by the Scientific and Ethics Committee of Andreas Syggros Hospital, and all participating individuals gave written informed consent before study participation.

### 2.2. DNA and RNA Extraction from FFPE Tissue

Tissues were microdissected with a syringe needle from 10 μm unstained FFPE sections. Dissections were supervised by a pathologist to maximise neoplastic cell content. Genomic DNA and RNA were isolated using the AllPrep DNA RNA FFPE kit (Qiagen, Hilden, Germany), according to the manufacturer’s suggestions with the modification that proteinase K digestion was performed at 56 °C overnight under stirring. Quantification of the samples was performed with a Qubit 4 Fluorometer (Thermo Fisher Scientific, Waltham, MA, USA) using a Qubit dsDNA HS Assay kit (Thermo Fisher Scientific, Waltham, MA, USA) for DNA and a Qubit RNA HS Assay kit (Thermo Fisher Scientific, Waltham, MA, USA) for RNA samples.

### 2.3. DNA Extraction from Blood Samples

Patients donated 2 mL of peripheral blood and DNA was obtained from peripheral lymphocytes of melanoma patients using the QIAamp DNA blood mini kit (Qiagen, Hilden, Germany). DNA concentration was quantified in samples prior to analysis using a Quant-iT dsDNA HS Assay kit (Thermo Fisher Scientific, Waltman, MA, USA).

### 2.4. Whole Exome Sequencing

Libraries were generated from a minimum of 200 ng DNA using a KAPA Hyper Prep Kit (Kapa Biosystems, Inc., Wilmington, MA, USA), targets were captured by SureSelect Human All Exon V6 (Agilent Technologies, Santa Clara, CA, USA), and sequencing was performed on a DNBseq-G400 sequencing platform at BGI TECH SOLUTIONS (Hong Kong).

### 2.5. Targeted RNA Sequencing

Transcriptomic analysis was performed using the AmpliSeqTM Transcriptome Human Gene Expression kit (Thermo Fisher Scientific, Waltham, MA, USA). Libraries were prepared following the manufacturer’s instructions with minor protocol modifications: 75–100 ng of RNA was used, 16 cycles of PCR were applied for cDNA amplification, and RNA was heated at 80 °C for 10 min before cDNA synthesis. Amplified cDNA libraries were evaluated for quality and quantified using the Tape Station 4150 instrument. Libraries were diluted to 100 pM, pooled, emulsion PCR-amplified and enriched as described in the Targeted DNA Sequencing section. Sequencing was performed on the Ion Torrent Proton™ sequencing system, using an Ion PI kit and PI chip V2 (8 libraries/chip).

### 2.6. Targeted DNA Sequencing and Data Analysis

The Oncomine™ melanoma panel (Thermo Fisher Scientific, Waltham, MA, USA), consisting of 2 primer pools targeting 439 and 440 amplicons (amplicon range: 125–175 bp) of 29 genes, was used for library preparation using 20–50 ng of FFPE DNA. We followed the manufacturer’s protocol with the modification that we increased the PCR cycles to 22 when the starting material was 20 ng. The libraries were quantified with a Tape Station 4150 (Agilent-G2992AA), diluted to 100 pM, pooled, and emulsion PCR-amplified with Ion PGM™ Hi-Q™ ion sphere particles (ISPs) using the Ion OneTouch™ 2 Instrument (Life Technologies; Thermo Fisher Scientific, Inc.) according to the manufacturer’s protocol. The template-positive ISPs were enriched using the Ion OneTouch™ ES instrument (Life Technologies; Thermo Fisher Scientific, Inc.) and sequenced with the Ion Torrent Proton™ sequencing system and the PI chip V2 (16 libraries per chip).

Sequence data were processed using the Torrent Suite 5.10.1 pipeline software, which is optimised for the Ion Torrent platform to perform raw data analysis and base calling, remove low-quality reads, and make alignments to the human genome (GRCh37/hg19). Variant calling was performed with Ion Reporter Server 5.18. The average total mapped reads per sample was 4.5 million. Variants supported only by reads in one strand or with an allele frequency <2% and an alternative allele count <10 were not included in the analysis.

### 2.7. Bioinformatics Analysis

#### 2.7.1. Whole Exome Sequencing Data Analysis

The analysis was based on a specially developed framework for NGS data analysis created for the project purposes, to identify SNPs/mutations and functions involved in cancer pathophysiology [38,39].

The first step for WES data analysis was to align the sequences on the reference genome (GRCh38) using BWA [40]. The sequences were then pre-processed using Picard, which identifies possible duplicates and reorders the sequences correctly. The next step was processing the reads using GATK (Genome Analysis Toolkit; version 4.1.9.0) [41,42,43,44] for quality control. Then, SNP calling was performed based on the reference genome (GATK HaplotypeCaller), and mutation calling based on the reference genome and a paired control sequence from the same patient (GATK MuTect2, [45]); blood samples were used as controls. Strand-specific artefacts, possibly due to DNA damage resulting from formalin fixation and storage time, were excluded from further analyses following the following rule: at least one incident per strand and over five incidents in both strands. Mutational signatures in the samples were uncovered using SigProfiler [46,47]. Funcotator from GATK was used for primary gene annotation.

#### 2.7.2. Somatic Variant Prioritisation and Pathway Analysis

OpenCRAVAT [48] was also used for integrative gene annotations of the vcf files for the incorporation of ancillary and up-to-date knowledge bases [49,50,51,52,53,54,55,56,57,58,59,60,61] for the functional prediction and clinical interpretation of mutations. In the case of coding somatic variants, these were graded using the Simple Variant Ranking Annotation Cancer Score tool [62]. Briefly, SVRACAS utilises various distinct annotation resources from OpenCRAVAT, aggregated in 4 major sources of evidence (clinical, variant effect prediction, cancer and expression) in order to assign each variant/gene a total ranking score in each sample. Next, subsequent functional analysis aimed to find cellular pathways and molecular mechanisms that are affected by the specific selected mutations in the previous step using BioInfoMiner [63]. vcfR [64] and GenVisR [65] packages were used for data manipulation and visualisation.

#### 2.7.3. Transcriptomic Data Analysis

Raw counts for each sample were downloaded from the ampliSeqRNA plugin of the Torrent Suite software. edgeR [66] was used for filtering of low count genes (≥1 cpm in over 2 samples) and normalisation, using the Trimmed Mean of M-values (TMM) between-sample normalisation method. Z-score values were calculated on normally distributed data by subtracting the overall average gene abundance from each gene’s expression and dividing by the standard deviation of all the measured counts across all samples.

For unsupervised clustering, a set of 1500 genes was used in log_2_CPM (counts-per-million) values, which were the most variably expressed (highest median absolute deviation, MAD) across 13 melanoma tumour samples. Consensus nonnegative matrix factorisation (CNMF) [67] clustering was implemented using the ExecuteCNMF() function of the CancerSubTypes R package [68], with 2- to 5-cluster-solutions and 1000 iterations. Silhouette plots and cophenetic correlations (for r = 2, 3, 4, 5, 6) were assessed to define the number of clusters in the optimal cluster solution (k = 3), defining three subtypes: cluster 1 (*n* = 2), cluster 2 (*n* = 4) and cluster 3 (*n* = 7). Each point on the cophenetic correlation graph was obtained from 50 runs of Brunet et al.’s algorithm [67].

Complete downstream RNA-seq analysis including filtering, normalisation and differential gene expression analysis between the MES-like and MEL-like melanoma tumour samples was performed in R (R version 4.2.1)/Bioconductor software (version 3.16), utilising the limma-voom pipeline [69,70]. At first, a non-specific count filtering was performed to exclude samples with zero count reads by keeping only genes expressed in at least one condition. With this step, 16,144 unique genes remained. Next, the TMM normalisation was implemented using the edgeR package (version 3.39.1), followed by the utilisation of the voom pipeline, to also incorporate sample weights and increase statistical power. Finally, the log_2_CPM values of the annotated genes for each sample were calculated, using the cpm() function of the edgeR package; a small prior.count value proportional to the library size was added to DGEList object to avoid errors by calculating the logarithmic value of zero. These normalised expression values were used for gene-level exploratory data analysis (EDA) with visual quality assessment providing insight into the possible relationships between the samples.

Gene set enrichment analysis (GSEA) [71] was performed twice, both on the gene expression dataset of normalised counts (log_2_CPM) in gct format and the preranked (based on log_2_FC) differentially expressed gene dataset derived from the MES- vs. MEL-like comparison. In the first case, the GSEA GenePattern module (version 20.4.0) [72] was utilised with the following parameters: 1000 permutations, collapse dataset to gene symbols—false, permutation type—gene set, enrichment statistic—weighted, metric for ranking genes—Signal2Noise, gene list sorting mode—real, gene list ordering mode—descending, max size of gene sets—500, and min size of gene sets—15. Fast pre-ranked gene set enrichment analysis (GSEA) was implemented with the fgsea R package [73] in the second case, by setting the estimated *p*-value (eps) argument to zero in order to estimate the *p*-value more accurately. As gene set database files in both GSEA analyses, we used the hallmark (h.all.v7.5.1.symbols.gmt) and Reactome (c2.cp.reactome.v2022.1.Hs.symbols.gmt) signature files in gmt format retrieved from the Molecular Signatures Database (https://www.gsea-msigdb.org/gsea/msigdb/, accessed on 16 September 2022 and 2 October 2022, respectively) [74], as well as the MITF and Invasive signatures of Hoek et al. [26]. The enrichment score (ES) reflects the degree to which a gene set-signature-pathway was overrepresented at the extremes (top-up-regulated or bottom-down-regulated) of the entire ranked list of differentially expressed genes (DEGs) by the comparison of MES-like vs. MEL-like melanoma tumours. The score was calculated by walking down the list of DE genes, increasing a running-sum statistic when we encountered a gene in our gene signature and decreasing it when we encountered genes not included in our gene signature. The magnitude of the increment depends on the correlation of the gene with the phenotype (MES-like or MEL-like). The enrichment score depicts the maximum deviation from zero encountered in the random walk. For the visualisation of the differentially altered pathways, customised bar plots were created using the R package ggplot2 (v.3.3.6).

Single-sample GSEA (ssGSEA) was implemented to clarify the degree to which the genes in a particular gene set were up- or down-regulated in a coordinated manner within each sample [71,75]. ssGSEA was performed based on the ssGSEA projection methodology described in Barbie et al. [75] by running the wrapper “ssgsea-gui.R” to ssGSEA script (https://github.com/broadinstitute/ssGSEA2.0, accessed on 29 June 2022). As gene set database files, we used the hallmark (h.all.v7.5.1.symbols.gmt) and Reactome (c2.cp.reactome.v2022.1.Hs.symbols.gmt) signature files in gmt format retrieved from the Molecular Signatures Database (https://www.gsea-msigdb.org/gsea/msigdb/, accessed on 29 June 2022) [74], as well as the MITF and Invasive signatures of Hoek et al. [26] reflecting the proliferative and invasive melanoma states. ssGSEA was performed on the normalised counts (log_2_CPM) of gene expression generated with the edgeR Bioconductor R package [66]. The script “ssgsea-gui.R” was sourced in R (version 4.2.1) running under Windows 10 64-bit (build 19044) by setting the following arguments: sample.norm.type: none; weight: 0.75; statistic: area.under.RES; output.score.type: NES; nperm: 1000; min.overlap: 10; correl.type: rank; run.parallel: TRUE. The normalised enriched scores (NES) of gene signatures were then median scaled between the 13 melanoma tumour samples and hierarchical clustering (ComplexHeatmap R package version 2.13.2) was conducted on rows using the Euclidean distance with minimisation of the total within-cluster variance using Ward’s minimum variance method (ward.D2).

#### 2.7.4. Correlation between BRAF and PPP6C Mutations in Public Datasets

Melanoma public data were downloaded from cBioPortal [76,77] for PanCancer TCGA and GENIE [78] studies. Statistical analyses were performed using R software (http://www.R-project.org/, version 4.2.1). Chi-square (χ2) hypothesis test of independence and Fisher’s Exact Test were performed to analyse the correlation between the frequency occurrence of *BRAF* (V600E or V600K) and *PPP6C* (R264C) mutation. Contingency tables of the distribution of *BRAF* mutation groups (V600E and V600K samples in rows) by the distribution of with and without the PPP6C R264C mutation groups (groups in columns) were analysed.

## 3. Results

### 3.1. Patient Characteristics and Type of NGS Analysis Performed

Forty-five patients diagnosed with CM in Andreas Syggros hospital between 2016 and 2020 participated in this study. The main clinical data of the patients are summarised in Table 1. One patient carried two melanoma lesions that were sequenced and analysed separately. Melanoma tissue from FFPE blocks was used to isolate DNA and RNA for molecular characterisation exploiting NGS. In the case where the isolated DNA of tumour was >200 ng and a blood sample of the patient was also available, WES was performed on both melanoma and blood DNA (*n* = 31 paired samples). For the patients for whom tumour DNA was limited, targeted sequencing using the Oncomine melanoma panel was performed instead; this is an amplicon-based methodology requiring much less input DNA (*n* = 16 samples; 2 samples were acquired from 1 patient). A subset of samples (*n*= 13) was also assessed by transcriptomic analysis.

### 3.2. Identification of Germ-Line Melanoma Risk Variants

Aiming to examine whether the patients had germline variations on possible melanoma susceptibility loci, WES data from blood DNA of 31 patients were analysed. We focused on a panel of SNPs associated with CM risk, which was derived from the GWAS catalogue database [79], enriched by putative melanoma risk SNPs based on the MelGene database [15,80]. The analysis was restricted to SNPs located in exon regions due to the nature of our data (WES data). The results are summarised in Figure 1, where the melanoma-risk associated alleles identified in the cases under study are shown, as well as their frequency in the Greek population [80] and the frequency in the European population as recorded in the GnomAD database [81]. The related SNPs include pigmentation-linked genes (*OCA2*, *SLC45A2*, *TYR*, *MC1R*), along with cell cycle associated and DNA repair genes (*ATM*, *CDKN2A*, *ERCC5*). Specific melanoma susceptibility alleles [82,83,84,85,86,87] were found in a number of patients.

### 3.3. Identification of Somatic Variants

Exome sequencing analysis of 31 paired—melanoma and blood—samples revealed a total number of 41,434 somatic variants in all patients. Regarding protein-altering variants, a total of 17,653 variants were identified, comprising Single Nucleotide Variants (SNVs), Multiple Nucleotide Variants (MNVs) and small insertions or deletions (InDels). Specifically, we identified 13,471 missense SNVs, 565 MNVs resulting in amino acid substitutions, 981 nonsense variants (948 SNVs and 33 MNVs), and 306 splice-site and 1792 frameshift variants. The complete list of somatic variants identified by WES is summarised in Appendix A. The number of somatic protein-altering variants ranged among the patients from 54 to 1965 and accordingly, the tumour mutation burden (TMB)—calculated as the rate of non-synonymous variations per Mb of the exome—was between 1.5 and 55 mutations/Mb.

Next, we analysed exome sequencing data to identify specific mutational signatures that could reflect the underlying mutational processes [20] (Figure 2). As expected, the majority of non-acral CMs were characterised by UV-related mutational signatures, which is typically observed in CM; additionally, a high rate of SBS5 signature was detected, which is characterised as a signature of unknown aetiology, but it has been correlated with age and smoking [20]. Two more mutational signatures were observed, namely SBS1, related to an endogenous mutational process such as spontaneous deamination of 5-methylcytosine, and SBS30, which has been associated with deficiency in base excision repair mechanisms.

Targeted sequencing of 29 genes was performed on 16 melanoma samples and revealed a total of 34 non-synonymous somatic mutations in 16 genes. Specifically, 28 missense, 5 nonsense and 1 splice-site mutations were identified. The complete list of the identified somatic variants identified by targeted sequencing is summarised in Appendix A.

To understand the significance of the identified mutations and mutated genes, we first compared our data with the COSMIC database [88,89]. We selected the top-20 mutated cancer-related genes (referred to as census genes) recorded in COSMIC for melanoma, as well as 19 more genes targeted by the Oncomine melanoma panel, resulting in a total of 39 genes. Figure 3 illustrates the distribution of somatic variants in these genes, where the information for 34 genes is shown because 5 genes did not carry any variant. Forty-four out of forty-six samples are depicted in the Oncoplot since in two samples no variant was detected in these genes. Clinical details for each sample are presented, including Breslow thickness, age group at diagnosis and melanoma subtype (SSM, LMM, ALM, IN SITU). A large proportion (16/23, 69.6%) of *BRAF*-mutated melanomas were in situ or thin melanomas (<1 mm Breslow) and the majority (5/8, 62.5%) of patients aged under 40 carried a BRAF V600E mutation. In contrast, V600K was found only in older patients and three out of four cases were located in the head and neck region. In addition, BRAF V600E carrying melanomas were mainly nevi-related as fourteen out of nineteen cases (73.7%) were raised on pre-existing nevi in areas with intermittent UVR exposure. A detailed description of clinicopathological data in the different mutational subtypes is summarised in Appendix A.

Regarding the prevalent melanoma-related mutated genes *BRAF*, *NRAS* and *NF1*, 38 known oncogenic mutations were found in all 46 melanoma samples. All of them were mutually exclusive, except one sample that carried an NRAS Q61K substitution along with an NF1 nonsense mutation. Regarding BRAF, twenty-three melanomas carried mutations at the hotspot position V600. Specifically, 18 melanomas carried the chr7:140753336 A>T SNV, resulting in a V600E substitution, one melanoma carried the MNV chr7:140753335 CA>TT, also resulting in a V600E substitution, and four melanoma samples carried the chr7:140753336 MNV AC>TT, which results in a V600K amino acid change. One sample was found to carry the BRAF V600E mutation co-occurring with a BRAF R347Q substitution. This specific variant has been previously reported in nevus sebaceous [90], a class of epidermal nevi, as well as in a few cases of non-skin cancers, as detected by searching the cBioPortal (https://www.cbioportal.org, accessed on 2 September 2022) and GDC Data Portal (https://portal.gdc.cancer.gov, accessed on 2 September 2022) platforms. Ten melanomas carried mutations in the Q61 position of NRAS (5 Q61K, 2 Q61L, 2Q61R and 1 Q61H) and one sample carried the rare, but likely, oncogenic substitution A59T [50,91]. Regarding *NF1*, three nonsense mutations were identified.

A number of melanoma recurrent mutations have been reported in the literature in several genes including *KIT*, *TERT*, *MAP2K1* and *MAP2K2*, *RAC* and *PPP6C* [92,93]. We searched in our data for such recurrent mutations that have been reported as clinically relevant. In particular, we exploited the OncoKB knowledgebase to identify oncogenic or likely oncogenic variants. In addition, we searched for non-hotspot, but putatively oncogenic, variants, such as truncating (nonsense or frameshift) mutations in well-established tumour suppressor genes (Figure 4). It is noteworthy that from the 46 melanomas under study, 43 samples carried at least one oncogenic or likely oncogenic somatic variant (Figure 3). Interestingly, two melanomas carried the PPP6C R264C variant, which has been reported to have a unique role in melanocytes and melanoma cells [94], co-occurring with the BRAF V600K substitution. This observation prompted us to search large cohorts for possible co-occurrence between BRAF V600K and PPP6C R264C. Data from two large-scale studies were used: the TCGA PanCancer Atlas for Skin Cutaneous Melanoma and the GENIE Cohort v12.0 for Melanoma. In both studies, PPP6C R264C co-occurred with both *BRAF* and *NRAS* mutations. Interestingly, we observed differences regarding the co-occurrence of PPP6C R264C with BRAF V600E and V600K. In the case of the TCGA Pan-Cancer Atlas (448 melanoma samples in total), among the 35 BRAF V600K mutated samples, 5 also had the R264C mutation in PPP6C (14.3%), while only 2/158 (1.3%) samples carried concomitant BRAF V600E and PPP6C R264C mutations (Fisher’s Exact Test—*p*-value = 0.00242). In agreement, regarding data of the GENIE project (2281 melanoma samples in total), among the 224 BRAF V600K mutated samples, 14 also carried the PPP6C R264C mutation (6.3%), while only 13/1269 (1%) samples carried concomitant BRAF V600E and PPP6C R264C mutations (Fisher’s Exact Test—*p*-value = 0.000006506) (Appendix A).

### 3.4. Functional Characterisation of Mutated Genes

In order to gain insights into the molecular mechanisms that are related with the genes carrying somatic variants, we performed functional enrichment analysis exploiting the BioInfoMiner platform (https://e-nios.com/products/bioinfominer/, accessed on 22 July 2022; [63]). Aiming to limit our analysis to genes carrying putatively oncogenic alterations, we prioritised the identified variants combining information from different levels, as described in Materials and Methods, resulting in a total of 1297 genes that represent candidate genes with altered function. Functional enrichment analysis revealed significantly enriched GO and Reactome terms encompassing key biological processes. The complete list of enriched terms is documented in Appendix A. Among the highest ranked of these, there were ontological terms related to extracellular matrix organisation, cell adhesion and cell–cell communication, developmental processes (highly represented by terms related to neurogenesis), axon guidance, signalling by Receptor Tyrosine Kinases (RTKs) and small GTPases, and angiogenesis (Figure 5).

### 3.5. Transcriptomic Profiling of Melanoma

Transcriptomic analysis was performed on total RNA isolated from 13 melanoma samples. Non-negative matrix factorisation (CNMF) [67] was implemented in order to distinguish potential molecular patterns for class discovery. CNMF of the top 1500 most variable genes based on their median absolute deviation (MAD) identified three main distinct clusters (Appendix A). Based on recent studies highlighting specific gene signatures that characterise melanoma cell states [23,31], we tested gene expression data from melanoma tumours for similarities with specific transcriptomic profiles of distinct melanoma phenotypic subtypes. In particular, we mapped the melanocytic-like (MEL) and mesenchymal-like (MES) gene sets reported by Andrews et al. [31] on our dataset. The MEL gene set consists of 94 genes and is characterised by the presence of classic melanocytic markers (e.g., *MITF*, *MLANA*, *TYR*, *DCT*, *SILV*, *OCA2*, *SOX10*), while the MES gene set consists of 149 genes and includes genes implicated in Epithelial to Mesenchymal Transition (EMT) (e.g., *ZEB1, AXL, ADAM12, COL1A1/5A1/6A2*). As reported [31], the MEL state is highly associated with the “melanocytic”/”proliferative” state, while the MES state is highly associated with the “undifferentiated”/“invasive” state and weakly with the “neural crest like” state. The relative expression of MEL and MES genes in the in-house primary cutaneous melanoma dataset (*n* = 13) is shown in Figure 6. Melanoma samples were separated into three distinct clusters. The first one (MEL-like) resembles the MEL cluster of Andrews et al., characterised by high expression of *MITF* and MITF targets and low expression of MES genes. The second cluster (MES-like) displays an opposite expression pattern, with low expression of MITF target genes and relatively high expression of MES-related genes. The third cluster shows mixed expression of MEL and MES gene sets, so its members could not be assigned to MEL or MES phenotypic classes; however, they probably represent transcriptional programs of phenotypes between melanocytic- proliferating and undifferentiated cells, such as the “hyperdifferentiated” state, the “intermediate” (“transitory-intermediate migrating”) state, the “starved” (“therapy-induced starved-like”) state and the NCSC-like state [23,27].

In order to evaluate the phenotypic diversity of our samples, we mapped the expression profile of key melanoma cell-state regulatory genes such as the *NGFR*, *RXRG*, *MITF*, *SOX10*, *SOX9* and *AXL* genes (Appendix A). The *SOX10* gene presented a similar expression profile with *MITF*, while *RXRG* had almost the same expression profile as *NGFR* across the samples. The *NGFR* and *RXRG* genes were upregulated in all non-MEL, non-MES samples, suggesting their stem-like characteristics, while they were found with low expression levels in MES- and MEL-like tumours, as expected. Principal Component Analysis (PCA) on the thirteen melanomasbased on the expression level of the aforementioned key melanoma cell-state regulatory genes (Appendix A) validated the separation of samples into three groups. The PCA loadings of *MITF* and *SOX10* contributed positively to the higher proportion of variance between the samples, explained by PC1, while the *AXL* contributed negatively. These loadings were mainly responsible for the separation of samples into MES-like and MEL-like. On the other hand, the *NGFR* and *RXRG* features contributed positively to the higher proportion of variance between the samples, explained by PC2, segregating significantly the MES- and MEL-like samples from all the others (Appendix A). Among the non-MEL, non-MES samples, two samples (17 and 50) were also fully concordant with the “neural-plastic” (NPLAS) subtype of Andrews et al., which is characterised by up-regulation of 78 and down-regulation of 6 genes (Appendix A). In contrast, sample 51 displayed the opposite expression pattern of the NPLAS gene set.

Although the previous analyses identified known markers of distinct melanoma phenotypes, we could not discriminate specific states other than the MEL and MES states, probably due to the mixed cell populations coexisting within a tumour. We next focused on MEL- and MES-like clusters and on the expression of representative genes for each one (Figure 7). Regarding the MEL-cluster, the selected genes were the *MITF* gene and its targets *TYRP, TYR, PMEL, MLNA, BCL2, BC2A1, PPARGC1A* and *CDK2* that were also found to be down-regulated in the MES-cluster (Figure 7A). On the other hand, the selected genes of the MES-cluster were the *ABCC3*, *ADAM19*, *ADAMTS6*, *AFAP1*, *AKR1C3*, *AOX1*, *AXL*, *CCL2*, *CFH*, *CLU*, *COL12A1*, *COL13A1*, *CSF1*, *DPYD*, *FBN1*, *VEGFC* and *ZEB1* genes that were found to be down-regulated in the MEL-cluster. (Figure 7B). Interestingly, all samples of the MEL-cluster are *NRAS* mutated, while all MES samples are *BRAF* mutated.

To further investigate the similarities and differences between the two principal melanoma clusters (MES- and MEL-like) at the functional level, we performed differential gene expression analysis, which revealed 1400 differentially expressed genes (DEGs) with |log2FC|>=2 and *p*-value < 0.05 (Appendix A). Among them, the *MITF* gene and its targets were found to be down-regulated in the MES-like group, while down-regulation was observed in the representative MES-like genes (Figure 7B), validating the results of melanoma cell state clustering (Figure 6). The functional enrichment analysis of differentially expressed genes via the BioInfominer platform revealed that the majority of up-regulated DEGs in the MES-like cluster is implicated mainly in the epidermal, epithelial cell and keratinocyte differentiation, such as the *SPRR1A* and *SPRR1B, TGM5, FLG, LCE* gene members, type II keratin-family genes, keratin-associated protein (KAP) family genes, *S100A7*, *S100A8*, *EREG* and many others, but also genes with a crucial role in cell-cell adhesion and extracellular matrix organisation (protocadherin gamma cluster genes, members of the S100 family, desmocollin and cadherin family genes, claudin family genes, and *NPNT* and *ICAM* gene members) and lipid transport (ATP binding cassette subfamily B and C genes, very low density lipoprotein receptor members, and solute carrier family genes) (Appendix A). Going one step further, we also conducted Gene Set Enrichment Analysis (GSEA), incorporating hallmark, Reactome and well-known melanoma published gene signatures (see section “Transcriptomic data analysis” in Materials and Methods). In Figure 8, selected gene sets manifesting different levels of enrichment between the MES- and MEL-like cell state conditions are highlighted. The top positively enriched gene sets in MES-like samples concern KRAS signalling, extracellular matrix organisation and the invasiveness of tumours coupled with an increased inflammatory response, while the top negatively enriched pathways include mechanisms such as DNA repair and protein folding, metabolic processes such as glycolysis and oxidative phosphorylation, and cell-cycle mitotic events that promote cell proliferation (Figure 8A,C). This kind of analysis also validated the behaviour of the MITF gene signature [26] with increased enrichment in MEL-like tumours (Figure 8B).

Finally, we carried out another systematic approach to test whether our tumour sample characterisation, which was based on clustering methods by profiling the MES- and MEL-like gene signatures, is also validated in single-sample GSEA (ssGSEA), further highlighting their heterogeneity at a functional level (Figure 9). This analysis supported the characterisation of MEL- and MES-like cases, as gene sets such as that of MITF by Hoek et al. were clearly positively enriched only in MEL-like samples. These samples were also characterised by enriched pathways including DNA repair, UV response and Myc-targets. Regarding the pathways enriched in MES-like samples, they include KRAS signalling, inflammatory response, Hoek et al.’s invasive signature and the epithelial to mesenchymal transition process. Sample 19 presents a deviation from the other two MES-like samples (3 and 18) concerning these positively enriched pathways, but as far as the pathways with relevantly low enrichment are concerned, it is clearly clustered with the MES-like phenotype (Figure 8). Regarding samples 17 and 50, which resembled the “neural-plastic” (NPLAS) subtype of Andrews et al. [31], they present a relatively high enrichment of pathways, including TGF-b signalling, UV response, P53 pathway and the invasive gene set of Hoek et al. [26], while processes related to cell cycle and proliferation are negatively enriched.

## 4. Discussion

In this study, we wanted to expand our previous work on CM [38] for the characterisation of somatic mutations and germline variants in patients with primary melanomas from Greece. This was accomplished not only by incorporating additional samples in our analyses, but also by broadening our approach towards other levels of molecular and phenotyping screening, and combining WES, targeted and transcriptomic analyses with clinical information.

The vast majority of analysed samples (43/46) was found to carry at least one clinically actionable somatic variant, involving not only the prevailing mutations in *BRAF*, *NRAS* and *NF1*, but also *KIT*, *MAP2K1* and *MAP2K2*, *RAC* and *PPP6C*. A disassociation trend between BRAF V600E and PPP6C R264C mutations was disclosed. PPP6C is a phosphatase recurrently mutated in melanoma and has been reported as a major MEK phosphatase in cells exhibiting oncogenic ERK pathway activation [95]. Concerning the association of specific mutations with clinical characteristics, a higher trend of BRAF V600E was linked to low Breslow thickness (<1 mm) or in situ melanomas. Melanomas that arise on sites with intermittent UVR exposure are more frequently of SSM type, more likely to harbour a BRAF V600E mutation and arise from a pre-existing nevus. They also occur more frequently in a younger population and on sites such as the abdomen, chest and back [96]. This case was verified in our study, with 44% of SSM carrying BRAF V600E mutations and 28% carrying *NRAS* mutations. A great proportion of BRAF V600E mutations (74%) were also identified in nevi-related melanomas and 68% of carriers of BRAF V600E had a melanoma in areas of low sun exposure, such as the trunk, and sun-protected areas of upper and lower extremities. Melanomas that arise on sites such as the head, with large amounts of cumulative UVR, more commonly occur in an older population, and five out of eight in our case were over 60 years old. Such melanomas are less likely to have a BRAF mutation (when present, they more frequently harbour a V600K than a V600E mutation) and have significantly higher mutation burden. They also show more frequent mutations of *NF1, NRAS* and non-V600E BRAF mutations. This mutation scenario was also proven in our cohort, with melanomas in face and scalp mostly carrying BRAF V600K, *NRAS* and *NF1* mutations, whereas only one out of eight patients (12.5%) carried the BRAF V600E mutation. In accordance with the published data, the three ALM melanomas in our cohort were not *BRAF* mutated; rather, they carried *NRAS*, *KIT* and *NF1* mutations. Moreover, the two ALM melanomas located on the foot showed no UV mutational signature, as expected, whereas the subungual melanoma had a UV mutation signature to a small extent, as has been recently observed [97].

A subset of samples was profiled at the transcriptomic level, and it was shown that specific melanoma phenotypic states could be inferred from bulk RNA isolated from FFPE primary melanoma tissue. Specific gene signatures that characterise melanoma cell states were explored, and two distinct clusters of patients, characterised by melanocytic- and mesenchymal-gene signatures, were highlighted. Accumulating evidence indicates that melanoma plasticity relies on a phenotypic switch between a proliferative/differentiated and an invasive/undifferentiated cell state [26,30]. Loss of MITF, the master regulator of melanocyte differentiation, and reprogramming of epithelial–mesenchymal transition (EMT)-inducing transcription factors, such as SNAIL, SLUG, ZEB1 and ZEB2, regulate the phenotype switching toward a MITF-low, invasive or neural crest stem cell-like phenotype [98]. In addition to transcription factors, a prominent role in melanoma phenotype switching is played by the CD271/NGFR neurotrophin receptor which promotes a stem-like and migratory phenotype [36]. Phenotype switching in melanoma is considered a key mechanism regulating not only invasion and metastasis, but also acquisition of resistance to targeted therapies and immunotherapy [99,100]. Thus, it is of paramount importance to efficiently characterise different melanoma states in primary tumours, as reflected in their altered transcriptomes, since they affect the crosstalk with the immune tumour microenvironment by possibly promoting immune escape [101]. Moreover, ssGSEA highlighted the activation of the EMT pathway in MES-like samples, in agreement with the upregulation of AXL, a receptor tyrosine kinase considered as an EMT marker [102]. In addition, the inflammatory response pathway was found to be enriched in MES samples as compared to MEL, accompanied by significantly high expression of *CXCL1* and its receptor, *CXCR2*, as well as chemokine receptors *CCR4* and *CCR6*, in melanomas characterised by the invasive gene signature. These molecules have been assigned a major role in growth, survival, motility, and invasion of human melanoma [103], while they may also trigger the infiltration of immunosuppressive cells, such as tumour-associated macrophages (TAMs) and cancer-associated fibroblasts (CAFs) into the tumour microenvironment [104,105]. Better understanding of how melanomas can exploit chemokine pathways could lead to the identification of new therapeutic interventions.

Another interesting finding of our research is that, the extracellular matrix (ECM), remodelling processes were found to be highly enriched, based on bioinformatics functional analysis, both in the list of prioritised genes carrying somatic variants, and in the DEGs derived from the comparison of MES- vs. MEL-like tumour samples. Since the crosstalk of cancer cells with their microenvironment—including stromal and immune cells, as well as the ECM—is a key player during tumour development, from initiation, growth and progression to metastasis [106], it is worth noting that relevant processes were highlighted in our analysis both at the mutational and gene expression level.

Through integrative bioinformatics analyses exploiting different levels of information, including well-established databases, state-of-the-art tools, functional and gene set enrichment analysis, we demonstrated molecular circuits that are implicated in melanoma cell programmes, shedding light on genes and pathways that support the molecular maintenance of phenotypic states. Our findings provide support of the melanoma cell state concept, along with genetic and transcriptomic programmes that remain highly conserved within melanoma tumours. Putative associations with specific immune features could influence the tumour microenvironment with a direct impact on melanoma progression and metastasis.

In the current research we tried to incorporate as many distinct levels of data as was feasible. Still, some limitations should be acknowledged. Presently, the assessment of clinically actionable mutations is performed on FFPE tumour biopsies; however, the amount of tumour tissue is often limited, and DNA quality may not be always optimal. For this reason, WES analyses were conducted for the detection of simple somatic mutations (SNVs and InDels) and not for the identification of copy number alterations, which has been shown to be an important genomic alteration during melanoma progression [107]. Another important restraint was the limited number of patients analysed through transcriptomics. To deal with this, a multi-level analysis was performed, exploiting the literature, databases, and advanced tools to incorporate information aiming at a better understanding of the underlying mechanisms of the identified different melanoma cell states.

Finally, our future goal is to further investigate the association of fundamental mechanisms involved in melanoma manifestation with the mutational landscape of tumours under the management of the immune system and its interplay with the core phenotypic melanoma cell states.

## 5. Conclusions

We present a comprehensive mapping of genomic mutational and transcriptional profile of patients with primary melanomas from Greece, through integration of multi-level molecular and phenotyping information, combining WES, targeted and transcriptomic analyses with clinical data. Our results revealed melanoma-associated susceptibility alleles, mainly in pigmentation-linked genes and DNA repair genes. Clinically actionable somatic variants were identified in 38 samples, along with functionally enriched processes perturbed from the mutated genes related to neurogenesis and developmental processes. Interestingly, we observed an increased co-occurrence of PPP6C R264C with BRAF V600K compared to V600E, which, to the best of our knowledge, has not been previously reported. Finally, despite the high level of heterogeneity, the systemic bioinformatics approach that we followed, based on the integration and correlation of different biologically informative layers, allowed us to evaluate the phenotypic melanoma state of tumours, highlighting transcriptional programmes and master regulators, such as *MITF*, *NGFR*, *AXL*, and *SOX10*, acting as driving forces that permit, or not, a cell population to switch from one state to another, defining tumour progression and aggressiveness with a direct impact on melanoma progression and anti-cancer drug resistance.

## Figures and Tables

**Figure 1 cancers-15-00815-f001:**
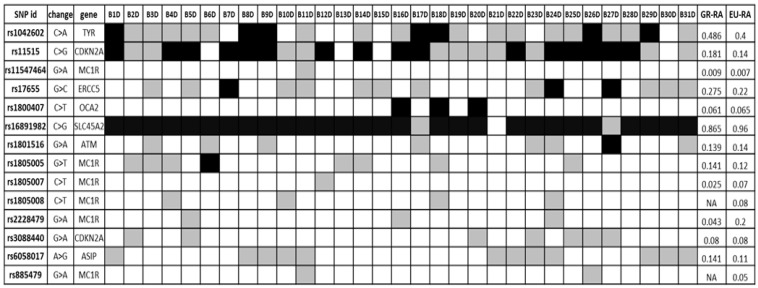
Germ-line SNPs putatively associated with melanoma risk; black = homozygous for melanoma risk-associated allele (RA), white = homozygous for reference allele, grey = heterozygous. GR-RA: Frequency of the risk allele in the Greek population, EU-RA: Frequency of the risk allele in the European population.

**Figure 2 cancers-15-00815-f002:**
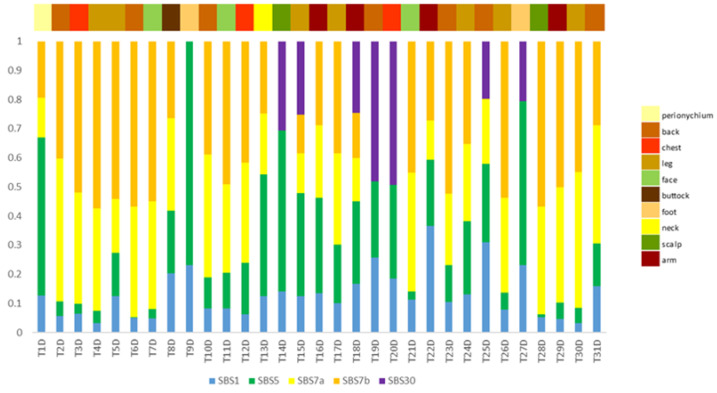
Mutational signatures per patient (*n* = 31). The identified signatures are colour-coded, and their proposed biological aetiology is as follows: SBS1: endogenous mutational process, spontaneous deamination of 5-methylcytosine; SBS5: unknown aetiology; SBS7a/b: exposure to ultraviolet light; SBS30: deficiency in base excision repair.

**Figure 3 cancers-15-00815-f003:**
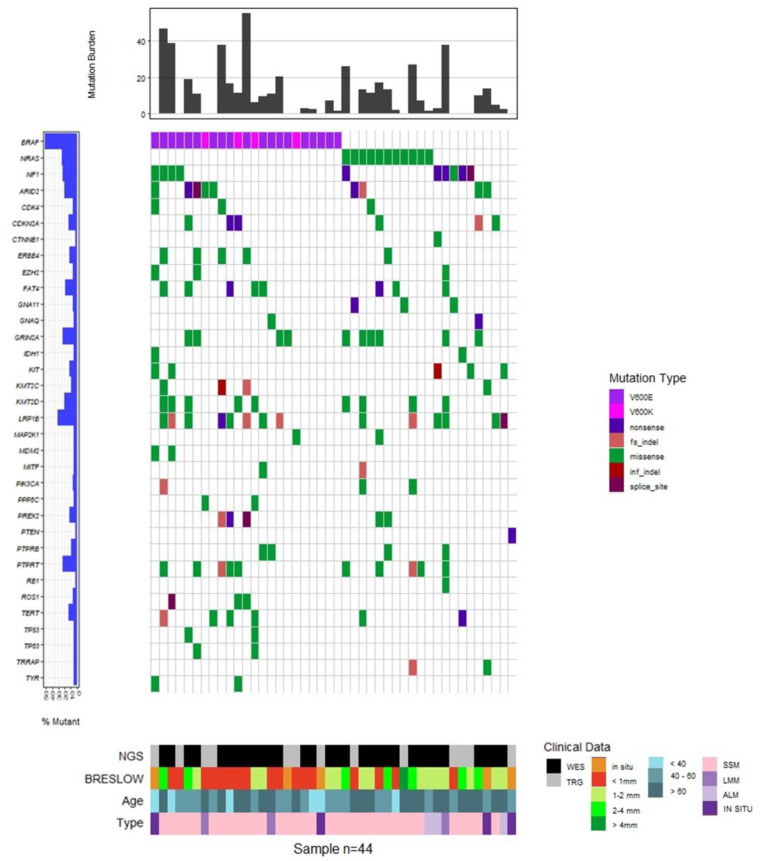
Oncoplot of somatic variance distribution in the melanoma patients under investigation. The top 20 cancer-related genes from COSMIC and additional 14 genes targeted by the Oncomine melanoma panel are shown (total = 34 genes), carrying at least one variant. The bar plot at the top of the figure displays the tumour mutation burden (TMB) for the samples analysed by WES. The observed frequency of mutations for each gene is presented at the left. Clinical data are shown: Breslow thickness, age group at melanoma diagnosis, and type of histological characterisation (SSM: Superficial Spreading Melanoma; ALM: Acral Lentiginous Melanoma; LMM: Lentigo Maligna Melanoma; IN SITU) and the sequencing technology applied (WES: Whole Exome Sequencing and TRG: Targeted Sequencing).

**Figure 4 cancers-15-00815-f004:**
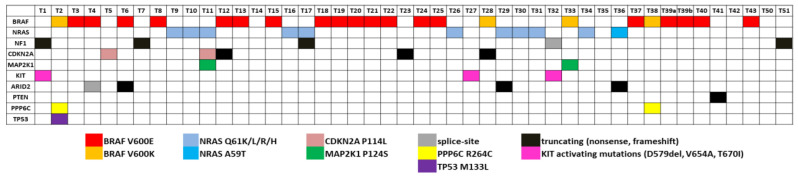
Somatic variants with clinical significance. The clinically relevant variants identified by WES or targeted sequencing in melanoma samples (*n* = 46) are shown. Different colours indicate specific variants, as shown at the bottom of the panel.

**Figure 5 cancers-15-00815-f005:**
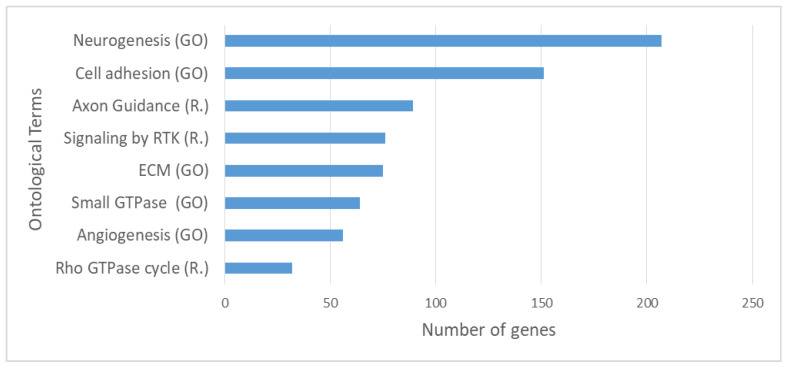
Top significantly enriched Gene Ontology (GO) and Reactome (R.) terms based on enrichment analysis on the 1297 prioritised genes carrying somatic variants with possible functional impact. RTK: Receptor Tyrosine Kinase, “Small GTPase” refers to the term: Signal transduction mediated by Small GTPase.

**Figure 6 cancers-15-00815-f006:**
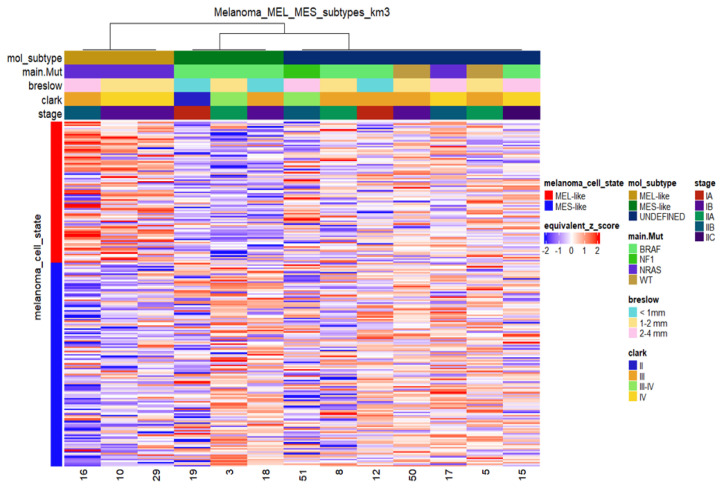
Heatmap of relative mRNA expression of MEL-like and MES-like associated genes in melanoma samples (*n* = 13) with sample cluster assignments by k-means clustering. K-means in columns (melanoma cases) identified three distinct groups based on the expression profile of the MEL-like (red vertical bar on the left) and MES-like gene sets (blue vertical bar on the left). Gene expression values were z-transformed and are coloured red for high expression and blue for low expression, as indicated in the scale bar. Clinical (Breslow; Clark and stage) and molecular information (mutated driver genes) of samples is added at the top as colour bars.

**Figure 7 cancers-15-00815-f007:**
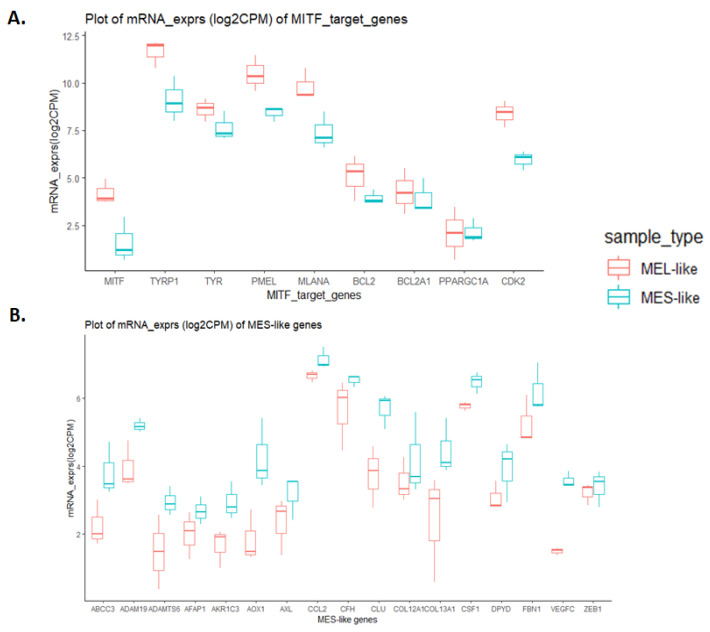
Boxplot diagrams of expression levels of classic melanocytic and mesenchymal gene markers in melanoma samples. Log2CPM mRNA expression values of MEL and MES groups were used. Boxplots indicate the median (thick bar), first and third quartiles (lower and upper bounds of the box, respectively), and lowest and highest data value within 1.5 times the interquartile range (lower and upper bounds of the whisker). Boxplot diagrams of expression levels of MITF target genes, highly expressed in MEL-like samples (**A**) and of mesenchymal gene markers, highly expressed in MES-like samples (**B**).

**Figure 8 cancers-15-00815-f008:**
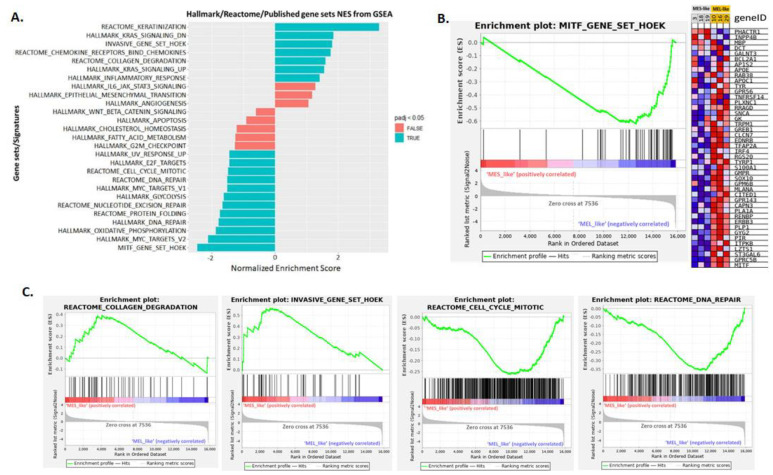
Gene set enrichment analysis (GSEA) results of MES-like versus MEL-like melanoma sample comparison. (**A**). Bar plot showing selected top enriched gene sets/pathways as derived from GSEA based on the differential expression analysis of MES-like versus MEL-like tumour melanoma samples. Bars in blue indicate significant enrichment at p-adj (FDR) <5%, bars in red represent gene sets with FDR >5% and a nominal *p*-value < 5%. A positive normalised enrichment score (NES) value indicates enrichment in the MES-like phenotype, a negative NES indicates enrichment in the MEL-like phenotype. (**B**). Gene set enrichment plot of MITF melanoma signature (Hoek et al. 2008), enriched in MEL-like vs. MES-like GSEA showing the profile of the running ES and positions of gene set members on the log2FC (Fold Change) rank-ordered list. The signal2Noise metric is graphed for each correlated gene in the ranked dataset. Heat map of 44 MITF signature genes (for each phenotype in the comparison of MES-like (grey column) vs. MEL-like (orange column). Normalised expression values are represented as colours and range from red (high expression), pink (moderate), light blue (low) to dark blue (lowest expression). (**C**). Four GSEA enrichment plots for representative molecular signatures (Cell-cycle, DNA-repair, Collagen-degradation, Tumour invasive signature) enriched in MES- and MEL-like tumours are shown. The top part of each GSEA plot shows the running enrichment score for validated genes specific for a particular pathway/signature as it moves down the ranked list. The bottom part of each plot shows the value of ranking matrices as it moves down the list of ranked genes. The red horizontal bar that terminates with a blue colour indicates a shift from positively correlated genes (red) to negatively correlated genes (blue). The *y*-axis represents enrichment score (ES) and the *x*-axis shows differential expressed genes (vertical black lines) represented in different pathways.

**Figure 9 cancers-15-00815-f009:**
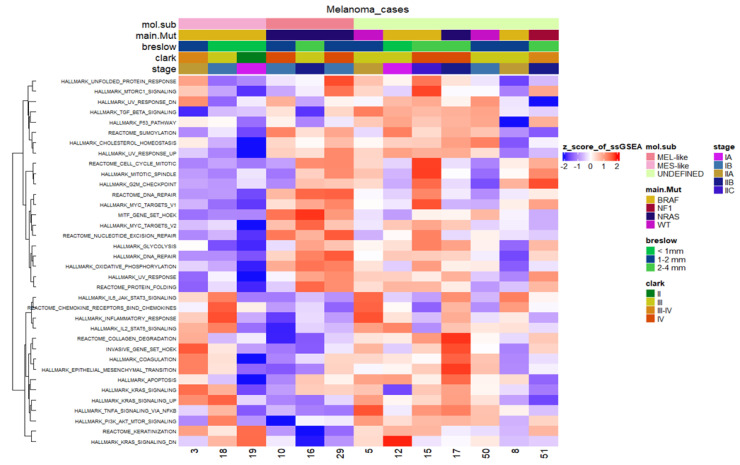
Heatmap of the scaled normalised enrichment scores (NES) of ssGSEA for selected hallmark/pathway gene sets. Rows are reordered by the method of hierarchical clustering (ward.D2 method, Euclidean distance), whereas columns are ordered based on the respective molecular subtype. Samples with relatively high enrichment of a given gene signature are marked in red and samples with relatively low enrichment are marked in blue. Clinical and molecular information (mutated driver genes) of samples is added at the top as colour bars.

**Table 1 cancers-15-00815-t001:** Basic clinicopathological characteristics of the 46 samples (45 patients); Histologic Type: SSM: Superficial Spreading Melanoma, ALM: Acral Lentiginous Melanoma, LMM: Lentigo Maligna Melanoma.

Clinicopathological Characteristics										
**Age**	Median age (range): 59 (29–81) yrs old									
**Sex**	Male 27 (58.7%)					Female 19 (41.3%)				
**Staging**	**0** 5 (10.9%)	**I** 27 (58.7%)			**II** 11(23.9%)		**III** 3 (6.5%)			**IV** 0 (0.0%)
**Breslow Thickness**	**0 mm** 5 (10.9%)	**<1 mm** 18 (39.1%)			**1–2 mm** 15 (32.6%)		**2–4 mm** 7 (15.2%)			**>4 mm** 1 (2.2%)
**Histologic Type**	**SSM** 36 (78.3%)		**ALM** 3 (6.5%)			**LMM** 3 (6.5%)			**IN SITU** 4 (8.7%)	
**Location**	**Head and Neck** 9 (19.6%)			**Trunk** 19 (41.3%)				**Extremities** 18 (39.1%)		

## Data Availability

The data that support the findings of this study are available on request from the corresponding author. The data are not publicly available due to privacy restrictions.

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
