# Peer review of "A Comprehensive Analysis of Cutaneous Melanoma Patients in Greece Based on Multi-Omic Data"

_cancers, 2023, doi:10.3390/cancers15030815_

Round 1
Reviewer 1 Report
A study by Kontogianni et al focuses on molecular characteristics of melanoma in Greek patients. The flow of the manuscript is correct, the data are clearly presented. However, the manuscript lacks novelty in general. The major conclusions are in line with general knowledge on melanoma, the genes mutated or diversely expressed are also known. Perhaps, the Authors could extend the study by providing additional validation of the results?
Specific comments:
1. In the introduction, CD271/NGFR should be mentioned as a part of currently acceptable phenotypes of melanoma.
2. Gene names should be written in italics.
Author Response
Point 1: However, the manuscript lacks novelty in general. The major conclusions are in line with general knowledge on melanoma, the genes mutated or diversely expressed are also known. Perhaps, the Authors could extend the study by providing additional validation of the results?
Response 1: We would like to thank the reviewer for the useful comments on our manuscript. Regarding transcriptomic analysis, we included several phrases at the Results section: page 12 lines 478-481, page 12 lines 487-492, page 13 lines 513-517 and a new paragraph page 12 lines 493-512. In addition we added a new supplementary Figure (Figure S3) aiming to extend our analysis (shown at the end of the revised manuscript, after the references section). We tried to overcome the limited number of samples of the transcriptomic analysis by in-depth bioinformatic analysis. Regarding the mutational characterisation of melanomas, the number of samples does not permit identification of new mutations. Nevertheles, the observed differences regarding the co-occurrence of PPP6C R264C with BRAF V600E and V600K, to the best of our knowledge, are reported for the first time.
Point 2: In the introduction, CD271/NGFR should be mentioned as a part of currently acceptable phenotypes of melanoma.
Response 2: In compliance with the reviewer’s comment we revised the introduction ( page 2 line 93- page 3 line 126).
Point 3: Gene names should be written in italics.
Response 3: Genes are written in italics, whereas proteins (genes with protein mutation) are presented with no formatting

Reviewer 2 Report
The authors characterize patients with primary cutaneous melanoma for both germline and somatic variations as well as alterations at the gene expression level, by applying whole exome, targeted and transcriptomic approaches and integrative, multi-layered bioinformatics analysis. The recent multi-omics analyzes (genomics, transcriptomics, proteomics, metabolomics, radiomics, etc.) and the technological evolution of data interpretation have allowed to identify and understand several processes underlying the biology of cancer. So the topic is relevant and also the article is well written.
There are minor issues that should be, where appropriate, improved, corrected or explained:
- The conclusions are too generic
- In the presentation of decimals, it should be unified throughout the text
- In line 118 it says that the median age is 58 years, however in table 1 it is 57.
- Table 1 includes patients with Breslow >4 mm, however in Figure 6 there are only patients up to 4 mm.
-The number of authors in the references is not presented homogeneously
Author Response
Point 1: - The conclusions are too generic
Response 1: We would like to thank the reviewer for the useful comments on our manuscript. The paragraph of conclusions has been rewritten (page 18 lines 709-724).
Point 2: In the presentation of decimals, it should be unified throughout the text.
Response 2: In compliance with the reviewer’s comment percentages are now given with 1 digit, only p-value statistics are presented with more.
Point 3: In line 118 it says that the median age is 58 years, however in table 1 it is 57.
Response 3: Thank you for pointing out this mistake. Median age was changed to 59 in the text & table, mistake due to samples deriving from the same patient.
Point 4: Table 1 includes patients with Breslow >4 mm, however in Figure 6 there are only patients up to 4 mm.
Response 4: Table 1 presents all samples (n=46), whereas figure 6 includes only samples undergone transcriptomic analysis (n=13); in those 13 samples there are only patients up to 4 mm.
Point 5: The number of authors in the references is not presented homogeneously
Response 5: We changed the bibliography style to MDPI style using Zotero.

Round 2
Reviewer 1 Report
My comments have been addressed.